# Assembly of a 3D Cobalt(II) Supramolecular Framework and Its Applications in Hydrofunctionalization of Ketones and Aldehydes

**Guoqi Zhang** [1,*], **Alex Wang** [1,†], **Haisu Zeng** [2], **Shengping Zheng** [2] and **Michelle C. Neary** [2]

1   Department of Sciences, John Jay College and PhD Program in Chemistry, The Graduate Center,
    The City University of New York, New York, NY 10019, USA; alexwang0317@gmail.com
2   Department of Chemistry, Hunter College, The City University of New York, New York, NY 10065, USA;
    hzeng@gradcenter.cuny.edu (H.Z.); szh0007@hunter.cuny.edu (S.Z.); mn1205@hunter.cuny.edu (M.C.N.)
*   Correspondence: guzhang@jjay.cuny.edu
†   Current address: Syosset High School, 70 S Woods Rd., Syosset, NY 11791, USA.

**Abstract:** A ditopic nitrogen ligand (*E*)-*N*′-(pyridin-4-ylmethylene)isonicotinohydrazide (**L**) containing both divergent pyridyl coordination sites and a hydrogen-bonding hydrazide–hydrazone moiety was synthesized. The Co(NCS)$_2$-mediated self-assembly of **L** has resulted in the synthesis of a novel 3-dimensional (3D) supramolecular framework (**1**) that features both coordination and hydrogen bonding interactions. X-ray structural analysis reveals the formation and coordination mode of **1** in the solid state. The rational utilization of coordination bonds and hydrogen bonding interactions is confirmed and responsible for constructing the 3D materials. Catalytic studies using **1** in the presence of an activator are performed for the hydroboration and hydrosilylation reactions of ketones and aldehydes, and the results are compared with previously reported cobalt-based polymeric catalysts.

**Keywords:** hydrazide–hydrazone ligand; Cobalt(II); 3D coordination network; hydrofunctionalization; ketones; catalysis



## 1. Introduction

The design and synthesis of metal–organic materials with well-defined structural topology and intriguing properties are attracting tremendous attention in recent years [1–3]. Among the driving forces responsible for the formation of novel materials between organic ligands and metal ions, the coordination and hydrogen-bonding interactions are considered to be the most important. In general, robust 1D or 2D networks can be first constructed by strong metal–organic coordination bonds and then higher dimensional structures will be assembled by hydrogen-bonding interactions [4]. Indeed, such a concept has been implemented in the construction of a number of functional metal–organic framework materials that have displayed fascinating properties in gas absorption/storage and catalysis in recent decades [5–8].

Metal-catalyzed hydroboration of unsaturated organic functionalities is one of the most attractive methods for the synthesis of alkylboronate esters that could be readily converted into the corresponding alcohols or used for C-C bond coupling reactions [9]. Although stoichiometric methods existed for a long time for the hydroboration of some functional groups such as carbonyls, nitrile, alkenes and alkynes by borohydride reagents, chemoselective hydroboration has been a challenge [10,11]. Except for the earlier contributions using noble metal catalysts, recent progress has focused on nonprecious, earth-abundant metals, such as Fe, Co, Ni, Cu, Zn, Mn, V, Al, Mg, etc., and thus far, numerous well-defined molecular complexes based on such base metals have been reported, including those showing good chemo- and regio-selectivity [9,12–24]. In addition, simplebase-catalyzed and non-catalytic hydroboration of carbonyl compounds have been recently disclosed [25,26]. It has been also reported that in the hydroboration of alkenes and alkynes some species of boranes and borohydrides have played a hidden role in the catalytic cycles [27].

We are particularly interested in the design and synthesis of base metal complexes using pincer-type PNP and NNN ligands and their applications in critical catalytic hydroboration reactions of alkenes, alkynes and carbonyl compounds [22–24]. Although superior catalytic performance using well-defined, discrete complexes of Co, Mn, V and Al as catalysts have been revealed, recent work has indicated that polymeric metal–organic materials assembled from ditopic pure-nitrogen ligands such as 4′-pyridyl-2,2′;6′,2″-terpyridine (tpy) and 4,2′;6′,4″-tpy upon coordinating with cobalt or iron salts can be used as highly efficient precatalysts for the hydroboration of various C = X (X = C, N or O) functionalities [28–31]. Significantly, it was revealed that the 1D polymeric chain assembled by 4′-pyridyl-2,2′;6′,2″-tpy and $CoCl_2$ was an excellent precatalyst for the hydroboration of various carbonyl compounds, alkenes and alkynes with high turnover efficiency and notable chemo- and regioselectivity in the presence of a base activator, potassium *tert*-butoxide (KO$^t$Bu) [28–30]. In addition, a 2D coordination network of $FeCl_2$ with a 4,2′;6′,4″-tpy derivative was found to catalyze the hydroboration of ketones and aldehydes in the air [31]. In general, such metal precatalysts utilize the rigid tpy ligands to form 1D polymeric chains or 2D coordination network, yet the synthesis of higher dimensional materials based on tpy-type ligands has been unsuccessful thus far in our group.

Intrigued by the fascinating catalytic properties of 3D metal–organic framework materials assembled by both coordination and hydrogen bonds, we continued on our catalyst design by moving from tpy ligands to semi-rigid Schiff-based ligands that contain both pyridine coordination sites and hydrogen bonding moiety. Thus, the structurally simple and synthetically facile ligand, **L** (Scheme 1), was designed in order to obtain novel metal–organic framework structures upon coordinating with octahedral metal ions such as cobalt(II). We have previously studied the fundamental coordination chemistry of similar ligands containing a hydrazide–hydrazone structural unit that can potentially form hydrogen bonding networks [32,33]. Herein, we report the synthesis and structural characterization of a 3D supramolecular framework (**1**) assembled by **L** and $Co(NCS)_2$ and its catalytic applications in chemoselective hydroboration and hydrosilylation of ketones and aldehydes.

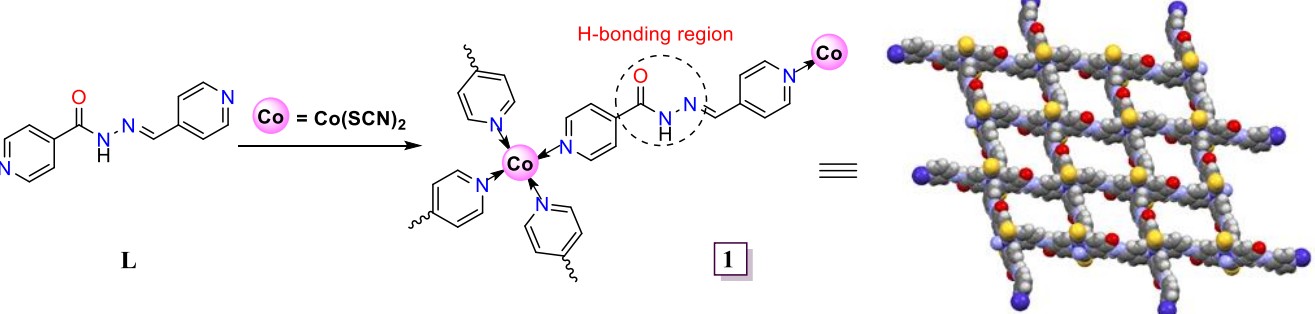

**Scheme 1.** Schematic representation of the assembly of 2D grid-like network found in **1** from ligand **L** and $Co(NCS)_2$.

## 2. Materials and Methods

### 2.1. Materials

Anhydrous grade solvents and liquid reagents used were obtained from Sigma-Aldrich (Milwaukee, WI, USA)and Fisher Scientific (Pittsburgh, PA, USA) and stored over 4 Å molecular sieves. FT-IR spectra were recorded on a Shimadzu 8400S instrument (Shimadzu Scientific Instruments, Columbia, SC, USA) with solid samples under $N_2$ using a Golden Gate ATR accessory. Elemental analysis was performed by the Midwest Microlab LLC in Indianapolis, IN, USA. Powder X-ray diffraction (PXRD) was performed with a Bruker D8 Discover microdiffractometer (Bruker Scientific LLC, Billerica, MA, USA) with the General Area Detector Diffraction System (GADDS) equipped with a VANTEC-2000 2D detector.

GC-MS analysis was obtained using a Shimadzu GCMS-QP2010S gas chromatograph mass spectrometer. Ligand **L** was prepared according to the literature [34,35].

### 2.2. Synthesis of **1**

A solution of (*E*)-*N'*-(pyridin-4-ylmethylene)isonicotinohydrazide (**L**, 113 mg, 0.5 mmol) in $CH_2Cl_2/MeOH$ (2:1, *v/v*, 15 mL) was placed in a test tube. MeOH (5.0 mL) was layered carefully on the top of the first solution, followed by a layer of a solution of $Co(SCN)_2$ (87.5 mg, 0.5 mmol) in MeOH (10 mL). The tube was capped and allowed to stand for one week at room temperature, during which period pale pink crystals suitable for single-crystal X-ray diffraction had formed on the walls of the tube and were isolated by decanting the solvent. The crystals were washed with MeOH and then dried in the air. Yield: 134 mg (86% based on **L**). FT-IR (solid, $cm^{-1}$): 3207 m, 2049 s, 1657 s, 1613 s, 1548 s, 1493 w, 1293 s, 1223 m, 1147 w, 1115 w, 1066 s, 1014 s, 948 m, 816 s, 657 s. Anal. Calc. for $C_{26}H_{20}CoN_{10}O_2S_2 \cdot CH_3OH$, C 49.17, H 3.67, N 21.24%; Found C 49.07, H 3.99, N 20.95%.

### 2.3. General Procedure for **1**-Catalyzed Hydroboration of Ketones and Aldehydes

In a glovebox under a nitrogen atmosphere, **1** (0.63 mg, 1.0 μmol, based on $Co(\mathbf{L})_2(NCS)_2$) and $KO^tBu$ (1.1 mg, 10 μmol) were placed in a 1.8 mL disposable vial equipped with a stir bar. Ketone or aldehyde (1.0 mmol) and pinacolborane (HBpin, 140.8 mg, 1.1 mmol) were then added. The reaction mixture was allowed to stir at room temperature for 2 h. At completion of the reaction, the reaction was exposed to the air and diethyl ether (10 mL) was added. The crude mixture was treated with 2N NaOH (1 mL) and 30% $H_2O_2$ (1 mL) stirred at room temperature for 1 h. The solution was extracted with ethyl acetate and washed with brine and water. The organic phase was concentrated under reduced pressure and then purified through a flash column chromatography with $SiO_2$ using ethyl acetate/hexane as an eluent. The products were characterized by $^1H$ and $^{13}C$ NMR spectroscopies.

### 2.4. General Procedure for **1**-Catalyzed Hydrosilylation of Ketones and Aldehydes

In a glovebox under a nitrogen atmosphere, **1** (0.63 mg, 1.0 μmol, based on $Co(\mathbf{L})_2(NCS)_2$) and $KO^tBu$ (1.1 mg, 10 μmol) were placed in a 1.8 mL disposable vial equipped with a stir bar. Ketone or aldehyde (1.0 mmol) and phenylsilane ($PhSiH_3$, 119.0 mg, 1.1 mmol) were then added. The reaction mixture was allowed to stir at room temperature for 2 h. At completion of the reaction, the reaction was exposed to the air and diethyl ether (10 mL) was added. The crude mixture was treated with 2N NaOH (1 mL) at room temperature for 2 h. The solution was extracted with ethyl acetate and washed with brine and water. The organic phase was concentrated under reduced pressure and then purified through a flash column chromatography with $SiO_2$ using ethyl acetate/hexane as an eluent. The products were characterized by $^1H$ and $^{13}C$ NMR spectroscopies.

### 2.5. General Procedure for **1**-Catalyzed Chemoselective Hydroboration

In a glovebox under a nitrogen atmosphere, **1** (0.63 mg, 1.0 μmol, based on $Co(\mathbf{L})_2(NCS)_2$) and $KO^tBu$ (1.1 mg, 10 μmol) were placed in a 1.8 mL disposable vial equipped with a stir bar. Ketone (1.0 mmol), another reducible substrate (1.0 mmol) and HBpin (128 mg, 1.0 mmol) were then added. The reaction mixture was allowed to stir at room temperature for 2 h. At completion of the reaction, the reaction was exposed to the air and the solvent was evaporated. The hydroborated product was analyzed by GC-MS using hexamethylbenezene as an internal standard to determine the yield.

### 2.6. X-ray Crystallography

A grey-pink crystal of **1** was mounted on a Cryoloop with Paratone-N oil and data were collected at 100 K with a Bruker APEX II CCD using Mo Kα radiation generated from a rotating anode. Data were corrected for absorption with SADABS (Version 2012/1) and the structure was solved by direct methods. All non-hydrogen atoms were refined anisotropically by full-matrix least-squares on $F^2$ with SHELXL (Version 2018/3) [36,37].

Hydrogen atom H3 on nitrogen atom N3 was found from a Fourier difference map. $U_{iso}(H)$ was set to 1.20 of $U_{eq}(N)$, and the N3-H3 distance was refined using a DFIX command set to 0.87(1) Å (final distance refined to 0.867(10) Å). All other hydrogen atoms were placed in calculated positions with appropriate riding parameters. The unit cell contained an unknown number of $CH_2Cl_2$ and/or MeOH molecules in two separate voids in the unit cell (each was 325 $Å^3$ with 75 electrons). These were treated as a diffuse contribution to the overall scattering without specific atom positions by SQUEEZE/PLATON [38,39]. The crystal structure was drawn with XP, and packing figures were drawn with the program Mercury v. 3.10.3. Crystallographic data for **1**: $C_{26}H_{20}CoN_{10}O_2S_2$, $M = 627.57$, grey-pink block, monoclinic, space group $P2_1/c$, $a = 8.1335(12)$, $b = 24.670(3)$, $c = 9.4320(13)$ Å, $β = 91.608(5)$, $U = 1891.8(4)$ $Å^3$, $Z = 2$, $Z' = 0.5$, $D_c = 1.102$ Mg $m^{-3}$, $μ$(Mo-K$α$) = 0.597 $mm^{-1}$, $λ = 0.71073$ Å, $T = 100(2)$ K. There were 22,705 measured, 3583 unique, and 2817 observed $[I > 2σ(I)]$ reflections. Refinement of reflections with $I > 2σ(I)$ (190 parameters, 1 restraint) converged at $R_1 = 0.0395$ ($R_1$ all data = 0.0572), $wR_2 = 0.858$ ($wR_2$ all data = 0.0919), GOF = 1.029. CCDC No. 1978581 contains the supplementary crystallographic data for this paper. These data can be obtained free of charge via https://www.ccdc.cam.ac.uk/structures/ (accessed on 25 March 2022), or from the Cambridge Crystallographic Data Centre, 12 Union Road, Cambridge CB2 1EZ, UK; Fax: (+44) 1223-336-033; or e-mail: deposit@ccdc.cam.ac.uk.

## 3. Results

### 3.1. Synthesis and Structural Characterization

The synthesis of ligand **L** was readily carried out in high yield by using a simple Schiff-based condensation between isonicotinoyl hydrazide and pyridine-4-carbaldehyde. Previously, this ligand has been utilized for the construction of mixed-ligand Zn and Cd metal–organic frameworks where the hydrazide–hydrazone moiety played an important role in fluorescent metal ion sensing and $CO_2$ absorption [34,35]. The cobalt-directed self-assembly of **L** with $Co(NCS)_2$ was conducted by a layering method in a $CH_2Cl_2$/MeOH solution. The block-like crystalline materials of **1** were obtained with an 86% yield after one week and the product was isolated by decanting the solvent and then washed with methanol and dried in the air. It was found that the large blocks of **1** tended to lose co-crystallized solvents readily upon exposure to the air and were cracked into microcrystalline materials. Single-crystal X-ray diffraction analysis was, therefore, carried out with a crystal collected rapidly from the solution and covered with paratone oil to avoid solvent loss from the lattice. Indeed, although its main structure could be unambiguously determined, there are a number of solvent molecules ($CH_2Cl_2$ and MeOH) in the unit cell that could not be well refined and hence treated with the program SQUEENE/PLATON [38,39]. The as-dried micro-crystalline sample was characterized by FT-IR and elemental analysis, as well as PXRD analysis. A comparison of the IR spectra between **L** and **1** revealed slight shifts of major absorptions upon coordination, except for the appearance of the characteristic absorption of the NCS group at 2053 $cm^{-1}$ (see Supplementary Material). Elemental analysis confirmed the composition of **1** with an empirical formula of $Co(L)_2(NCS)_2$ containing one molecule of MeOH in each repeating unit. The PXRD measurement of the as-dried crystals revealed the partial loss of its crystallinity and a pattern distinct from that simulated from single-crystal diffraction data, indicating the possibility of a phase transition due to the partial loss of co-crystallized solvents, which are frequently observed in porous metal–organic framework [40–42].

In our strategy, the octahedron-coordinating cobalt(II) ions are expected to bind with the terminal pyridyl-N atoms to form a 2D supramolecular architecture, while leaving the bridging hydrazide–hydrazone area available for the further assembly by 2D coordination sheets via hydrogen bonding interactions (Scheme 1). Accordingly, X-ray structural analysis of **1** confirms that a novel 3D framework is constructed through 2D sheet-like assembly by coordination bonds between **L** as linkers and octahedral $CoN_4(NCS)_2$ as the nodes followed by anticipated hydrogen bonds of the hydrazide–hydrazone regions.

1 crystallizes in the monoclinic space group *P*2(1)/*c* and its ORTEP structure showing the metal coordination environment is represented in Figure 1. The asymmetric unit contains 1 independent ligand molecule, 0.5 cobalt centers and 1 thiocyanate ion. Structural expansion around the metal center leads to the formation of a hexa-coordinate $Co^{II}$ center. Each $Co^{II}$ is surrounded by four ligand molecules to form a $CoN_6$ sphere, and the resultant $Co-N_{pyridyl}$ distances are 2.1760(18) and 2.2203(17) Å, while the $Co-N_{NCS}$ bond is slightly shorter (2.058(2) Å). The adjacent N-Co-N angles are in the range between 87.04(7) and 92.96(7)°, with the cobalt ion residing in an almost perfect octahedral coordination environment (Figure 1). Further inspection of the structure found that each ligand bonds to two cobalt centers by the end pyridyl sites, thus linking the metal centers to form a 2D sheet-like network that contains large grids (Figure 2a). In each grid structure, the adjacent Co···Co distance is 15.440 Å, while the diagonal Co···Co distances are 24.670 and 18.573 Å, respectively, due to the slight bending conformation of **L**. Interestingly, the 2D network was further assembled to a 3D architecture through inter-layer hydrogen bonding interactions using the hydrazide-NH as an H-bonding donor and C = O as the H-bonding acceptor (N···O = 2.827(2) Å, < N-H···O = 167(2)°), as shown in Figure 2b. Such a hydrogen bonding packing mode using the central zone of the ligands results in a more condensed 3D molecular stacking, while forming small channels in the crystals with a dimension of ca. 7 × 7 Å, which are mainly occupied by diffused solvents that rapidly evaporate in the air and can be further removed under vacuum.

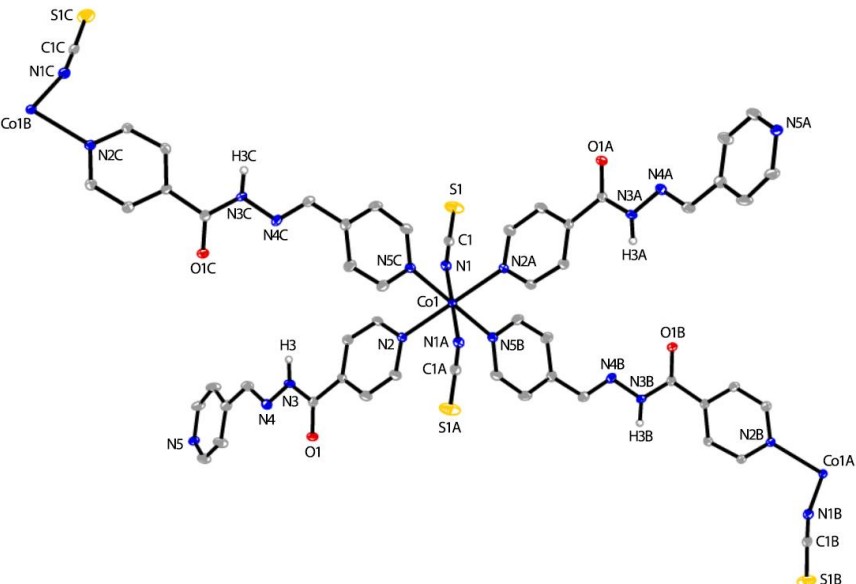

**Figure 1.** The ORTEP representation of **1** with thermal ellipsoids plotted at the 30% probability level. Hydrogen atoms bound to carbon are omitted for clarity.

### 3.2. Catalytic Applications

Next, we evaluated the catalytic performance of the 3D material **1** as a precatalyst for the hydroboration of ketone and aldehyde. First, acetophenone was chosen as a model substrate to react with the hydride source, pinacolborane (HBpin), in the presence of **1** and a base additive, the conditions adopted from the 1D Co coordination polymer ($[Co(4-pytpy)Cl_2]_n$)-catalyzed hydroboration as previously reported [28]. The results for the screening of reaction conditions are summarized in Table 1. It was revealed that the reaction of acetophenone with HBpin (1.1 eq.) did not proceed in the presence of **1** (0.1 mol% based on one $Co(\mathbf{L})_2(NCS)_2$ unit) without the addition of an activator (entry 1). Likewise, the reaction gave only a trace amount of product in the presence of $KO^tBu$ (1 mol%) alone as detected by GC-MS analysis (entry 2). In contrast, when the $Co(NCS)_2$ salt combined with $KO^tBu$ was used, a moderate yield was observed (entry 3). The combination of **1** and

KO$^t$Bu was, therefore, crucial for the formation of active catalytic species and to this end, a quantitative yield of the desired boronate ester was obtained in THF at room temperature (entry 4). The results are identical to those obtained with [Co(4-pytpy)Cl$_2$]$_n$ as a precatalyst (entry 5). The reactivity of the present catalyst is comparable to the reported 1D cobalt coordination polymer. We further studied the effect of different solvents on this reaction (entries 6–9). It was found that although benzene and toluene were less ideal solvents, pentane and diethyl ether are viable solvents that rendered the reaction with a slightly lower yield. It was also delightfully observed that the reaction can proceed equally well without the use of a solvent (entry 10), indicating the promising applications of this catalyst in green chemistry [13,31]. We also investigated the effect of additives as it is necessary for the present catalysis. Replacing KO$^t$Bu with KOH, K$_2$CO$_3$, NaBF$_4$ or NaBH$_4$ generally resulted in lower yields of the boronate ester (entries 11–14). However, the strong base, lithium bis(trifluoromethane)sulfonimidate (LiNTf$_2$), was found to facilitate the reaction with comparable results (entry 15).

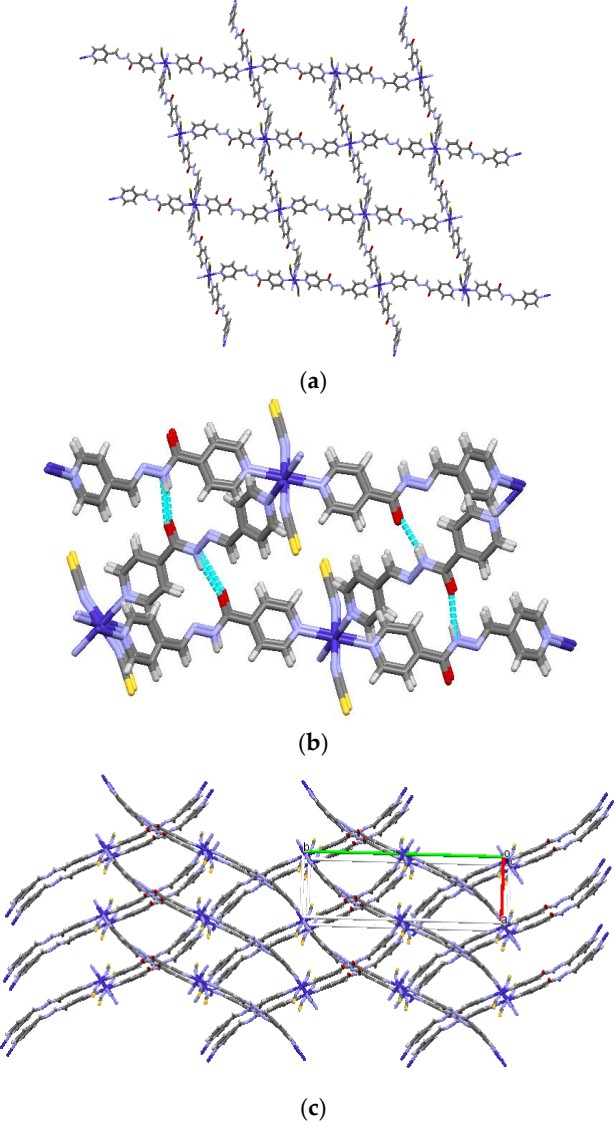

(**a**)

(**b**)

(**c**)

**Figure 2.** (**a**) The capped-stick representation of the partial 2D sheet-like coordination network. (**b**) The local ball-stick representation showing the interlayer N-H···O hydrogen bonding interactions drawn in green dashed lines. The grid in mid-layer is shown in blue. (**c**) The The 3D packing mode of **1** along the crystallographic c axis showing the void molecular channels (diffused solvents are removed for clarification).

**Table 1.** Catalytic test for **1**-catalyzed hydroboration of acetophenone with HBpin [a].

| Entry | Catalyst | Activator | Solvent | Yield/% [b] |
|---|---|---|---|---|
| 1 | **1** | - | THF | 0 |
| 2 | none | KO$^t$Bu | THF | <5 |
| 3 | Co(NCS)$_2$ | KO$^t$Bu | THF | 68 |
| 4 | **1** | KO$^t$Bu | THF | 99 |
| 5 | [Co(4-pytpy)Cl$_2$]$_n$ | KO$^t$Bu | THF | 99 |
| 6 | **1** | KO$^t$Bu | Et$_2$O | 95 |
| 7 | **1** | KO$^t$Bu | toluene | 80 |
| 8 | **1** | KO$^t$Bu | benzene | 86 |
| 9 | **1** | KO$^t$Bu | pentane | 95 |
| 10 | **1** | KO$^t$Bu | none | 99 |
| 11 | **1** | KOH | none | 58 |
| 12 | **1** | K$_2$CO$_3$ | none | 67 |
| 13 | **1** | NaBH$_4$ | none | 85 |
| 14 | **1** | NaBF$_4$ | none | 18 |
| 15 | **1** | LiNTf$_2$ | none | 96 |

[a] Conditions: acetophenone (1.0 mmol), HBpin (1.1 mmol, 1.1 eq.), **1** (0.1 mol%), activator (1 mol%) and solvent (0.5 mL), rt, 2 h. [b] Determined by GC-MS analysis with hexamethylbenzene as an internal standard.

Encouraged by the high catalytic activity found for the combined catalyst of **1**/KO$^t$Bu under neat and mild conditions, we explored its application for a range of ketones and aldehydes. The results are shown in Table 2. Acetophenone bearing electro-withdrawing and -donating groups, or the bulkier phenyl isopropyl ketone, furnished the reaction with high yields of the corresponding boronate esters under neat conditions and secondary alcohols could be readily obtained by hydrolysis (entries 2–5). In addition, α, β-unsaturated ketone and heterocyclic ketone are both suitable substrates for hydroboration, affording the corresponding alcohols with 90% and 82% yields, respectively (entries 6 and 7). Furthermore, aldehydes are also found to be hydroborated effectively in the presence of **1**/KO$^t$Bu under neat conditions (entries 8–10). Primary alcohols could be isolated upon hydrolysis of the boronate esters with 86–90% yields. We are also interested to see whether other hydride sources can be used for the reduction of ketones and aldehydes. Thus, the popularly used hydride, phenylsilane (PhSiH$_3$), was tested for the relevant hydrosilylation catalysis. It was delightful to find that both ketones and aldehydes proceed well for hydrosilylation with PhSiH$_3$ under neat conditions, and the corresponding alcohol products were isolated in 84–92% yields after hydrolysis (entries 11–15). Finally, we tried to expand the applications of this catalyst system for the hydroboration or hydrosilylation of alkenes and alkynes. Unfortunately, the results reveal that the present catalyst is almost inactive for such substrates for both types of hydrofunctionalizations (entries 16–18). This is, however, in contrast with the previously reported tpy-based cobalt(II) coordination polymer which is a highly efficient catalyst for the regioselective hydroboration of both polar and non-polar unsaturated bonds [28,29]. These results indicate the influence of the ligand on the catalytic activity/selectivity. Despite the polymeric nature of **1**, the catalyst system appeared to be a homogeneous solution under the catalytic conditions (in the presence of KO$^t$Bu and HBpin) and we were unable to recycle the catalyst as a solid by filtration or centrifugation of the reaction mixture, indicating the polymeric structure might have decomposed. At this point, the hidden role played by a borane formed in the reaction mixture and borohydride could not be excluded [27].

**Table 2.** Catalytic hydroboration or hydrosilylation of ketones and aldehydes using **1** as a precatalyst [a].

| Entry | Substrate | Hydride | Product | Yield (%) [b] |
|---|---|---|---|---|
| 1 | | HBpin | | 92 |
| 2 | | HBpin | | 94 |
| 3 | | HBpin | | 92 |
| 4 | | HBpin | | 95 |
| 5 | | HBpin | | 89 |
| 6 | | HBpin | | 90 |
| 7 | | HBpin | | 82 |
| 8 | | HBpin | | 88 |
| 9 | | HBpin | | 90 |
| 10 | | HBpin | | 86 |
| 11 | | PhSiH$_3$ | | 92 |
| 12 | | PhSiH$_3$ | | 90 |
| 13 | | PhSiH$_3$ | | 84 |
| 14 | | PhSiH$_3$ | | 85 |
| 15 | | PhSiH$_3$ | | 90 |

**Table 2.** *Cont.*

| Entry | Substrate | Hydride | Product | Yield (%) [b] |
|:---:|:---:|:---:|:---:|:---:|
| 16 | | HBpin | | 5 [c] |
| 17 | | HBpin | | 3 [c] |
| 18 | | PhSiH$_3$ | | trace [c] |

[a] Conditions: substrate (1.0 mmol), HBpin or PhSiH$_3$ (1.1 mmol, 1.1 eq.), **1** (0.1 mol%), KO$^t$Bu (1 mol%), neat, rt, 2 h. [b] Isolated yields. [c] Yields determined by GC analysis.

### 3.3. Chemoselectivity

Chemoselectivity is a critical issue when a catalyst is considered to be used in a practical process. To establish the chemoselectivity of the present catalyst for ketone and other reducible chemicals, we conducted several additional experiments to evaluate the present catalyst. Thus, several parallel reactions were set up while using two mixed substrates under the standard catalytic conditions as shown in Scheme 2. First, equimolar acetophenone and benzaldehyde were used to react with 1 equiv. of HBpin in the presence of **1**/KO$^t$Bu under neat conditions, and it was found that hydroboration selectively occurred in the aldehyde with a 90% yield, and the ketone was fully recovered. Likewise, competing experiments using equimolar acetophenone and other reducible chemicals such as alkene, alkyne, nitrile, ester and amide were performed. The results reveal that in all cases, the hydroboration was chemoselective for ketone over other functional groups. Interestingly, while the presence of other reducible substrates affected the reaction of ketone hydroboration with lower yields, the tertiary amide was found not to interrupt the reaction at all (Scheme 2).

**Scheme 2.** Chemoselective hydroboration between ketone and other reducible substrates. Conditions: acetophenone (1.0 mmol), reducible substrate (1.0 mmol), **1** (0.1 mol%), KO$^t$Bu (1 mol%) and HBpin (1.0 mmol), neat, rt, 2 h.

## 4. Conclusions

In conclusion, we have synthesized and characterized a novel supramolecular framework (**1**) by the self-assembly of a pyridine-based hydrazide–hydrazone ligand (**L**) and Co(NCS)$_2$. X-ray structural analysis has determined the molecular structure to be a 3D framework resulting from the hydrogen-bonding packing between 2D sheets of the extended coordination between **L** and metal ions. The catalytic applications of the as-synthesized polymeric materials in combination with an activator for hydroboration (and hydrosilylation) reactions of ketones and aldehydes were explored. The results suggest that **1** can be employed for both hydroboration and hydrosilylation catalysis for ketones and aldehydes under mild conditions, although it is not a viable catalyst for alkene and alkyne hydrofunctionalization. Good chemoselectivity of this catalyst system was also observed for aldehyde over ketone as well as ketone over other reducible groups such as alkene, alkyne, nitrile, ester and amide.

**Supplementary Materials:** The following supporting information can be downloaded at: https://www.mdpi.com/article/10.3390/chemistry4020029/s1, Supplementary data (catalytic details and NMR spectroscopy of isolated products). Crystallographic data for this paper have been deposited with the CCDC, accession numbers 1978581. These data can be obtained free of charge via www.ccdc.cam.ac.uk/data_request/cif (accessed on 25 March 2022) or by emailing data_request@ccdc.cam.ac.uk. Reference [43] is cited in the supplementary materials.

**Author Contributions:** Conceptualization, G.Z. and S.Z.; synthesis, characterization and catalysis, G.Z., A.W. and H.Z.; crystallographic analysis, M.C.N.; writing—original draft preparation, G.Z.; writing—review and editing, G.Z., A.W., S.Z. and M.C.N.; supervision, G.Z.; project administration, G.Z.; funding acquisition, G.Z. All authors have read and agreed to the published version of the manuscript.

**Funding:** This research was funded by the National Science Foundation, grant number CHE-1900500 and the City University of New York PSC-CUNY awards, grant numbers 63809-0051 and 64254-0052.

**Data Availability Statement:** Data is contained within the article or Supplementary Materials.

**Conflicts of Interest:** The authors declare no conflict of interest. The funders had no role in the design of the study; in the collection, analyses, or interpretation of data; in the writing of the manuscript, or in the decision to publish the results.

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
