# Peer review of "Assembly of a 3D Cobalt(II) Supramolecular Framework and Its Applications in Hydrofunctionalization of Ketones and Aldehydes"

_chemistry, doi:10.3390/chemistry4020029_

Round 1

Reviewer 1 Report

Guoqi et al. reported a 2D cobalt(II) cordination polymer for the hydrofunctionalization of ketones and aldehydes. The work is interesting. I suggest the acceptance of the paper after a few changes.

  1. How about the reusable of the 2D cobalt(II) cordination polymer in the hydrofunctionalization reaction, as listed in table 1.

2. Pls compare the XRD of the 2D cobalt(II) cordination polymer after the catalyses.

3. pls briefly discuss the mechanism of the hydrofunctionalization over cobalt(II) cordination polymers

Author Response

Responses to reviewer:

Point 1.  How about the reusable of the 2D cobalt(II) cordination polymer in the hydrofunctionalization reaction, as listed in table 1.

Response 1: Thanks for the comments. As stated in the catalytic part of the main text, the polymeric structure appeared to dissolve in the presence of KOtBu and HBpin through the possible formation of soluble oligomers, the catalyst could not be recycled after the catalysis.

Point 2.  Pls compare the XRD of the 2D cobalt(II) cordination polymer after the catalyses.

Response 2: For the same reason as above mentioned, the polymer materials was not recyclable after the catalysis and hence we could not obtain the XRD after the catalysis.

Point 3.  pls briefly discuss the mechanism of the hydrofunctionalization over cobalt(II) cordination polymers

 Response 3: Thanks for the helpful suggestion. Considering the complexity of the reaction mixture, in particular the fact that the polymeric structure may not have maintained during the catalytic cycle, we felt it was immature to draw a conclusion on the plausible mechanism at this stage. However, because of the high catalytic efficiency the materials has displayed we are further exploring the catalytic process by isolating and characterizing any active (soluble) species as key intermediates. This part of work is currently in progress and would be published in due course when we have a full understanding of the mechanism.

Reviewer 2 Report

Apart from the problems identified by the authors, the crystal structure solution and refinement is concerning. The number of reflections used for structure determination seems very low, 3583, and is fewer than the number used for the unit cell determination (7167). I would expect a number around quadruple that number for such a structure. The zero value for R(equivalents) is a sign of this: all the reflections are unique. This appears to be a consequnce of the minima of both k and l being 0 rather than -30 and -11 respectively, and thus only one quarter of the expected reflections being measured. The hkl file deposited with the CCDC supports this. This in turn may be a consequence of the unit cell initially appearing to be orthogonal (beta is cloe to 90 degrees). Nevertheless it is remiss of the crystallographers not to have collected all the reflections. It is my opinion that at present the crystal structure should not be published. I recommend that the authors ask an experienced crystallographer to look over their crystal data and help them rectify the problem, which may mean re-collecting the data. 

Additionally: The experimental sections states that there were 3583 reflections and 2778 unique ones. This should be 3583 unique reflections of which 2778 are observed (I  > 2sigma(I)). The experimental section also states that H(3) was refined with a N(3)-H(3) distance of 0.87(0.01) Angstroms (which should be written as 0.87(1) Angstroms), but the cif gives the distance as 0.859(9) Angstroms. On page 5 it is stated that the asymmetric unit contains one cobalt center, but this is really a 1/2 cobalt atom because Z' = 0.5. (It would be helpful if Z' were given along with Z in the experimental details.) 

Author Response

Responses to reviewer:

Point 1.  Apart from the problems identified by the authors, the crystal structure solution and refinement is concerning. The number of reflections used for structure determination seems very low, 3583, and is fewer than the number used for the unit cell determination (7167). I would expect a number around quadruple that number for such a structure. The zero value for R(equivalents) is a sign of this: all the reflections are unique. This appears to be a consequnce of the minima of both k and l being 0 rather than -30 and -11 respectively, and thus only one quarter of the expected reflections being measured. The hkl file deposited with the CCDC supports this. This in turn may be a consequence of the unit cell initially appearing to be orthogonal (beta is cloe to 90 degrees). Nevertheless it is remiss of the crystallographers not to have collected all the reflections. It is my opinion that at present the crystal structure should not be published. I recommend that the authors ask an experienced crystallographer to look over their crystal data and help them rectify the problem, which may mean re-collecting the data. 

Response 1: Thank you for pointing out the issues we have overlooked before the submission of the manuscript. During the revision, we were able to get kind help from an experienced crystallographer, Dr. Michelle Neary, whose contribution for the work is now recognized as a co-author of the revised article. Here is the response from Dr. Neary.

Upon reviewing the data, it become apparent that something happened to the .hkl file when SQUEEZE was applied. Somehow, the file was transformed such that only the unique reflections remained. It turns out that the original .p4p and pre-SQUEEZE .hkl (with 22705 total reflections) were provided by the original crystallographer, so the data were reprocessed. SQUEEZE was employed again, but this time, all the original reflections were retained in the .hkl, and R(equivalents) is now a much more reasonable 0.0512. This time, no solvent molecules were explicitly added back into the atom list since we cannot know with confidence whether the solvent in the lattice was MeOH, CH2Cl2, a mix of the two, or how many molecules there were.

There were very minor changes to the bond lengths and other parameters (which have been accordingly modified in the results section), but the overall geometry remains essentially the same, supporting the prior analysis.

Point 2.  Additionally: The experimental sections states that there were 3583 reflections and 2778 unique ones. This should be 3583 unique reflections of which 2778 are observed (I  > 2sigma(I)).

Response 2: The experimental section now reads “There were 22705 measured, 3583 unique, and 2817 observed [I >2σ(I)] reflections.” to clarify all three reflection values.

Point 3.  The experimental section also states that H(3) was refined with a N(3)-H(3) distance of 0.87(0.01) Angstroms (which should be written as 0.87(1) Angstroms), but the cif gives the distance as 0.859(9) Angstroms.

Response 3: It appears that in the original analysis, a DFIX command of 0.87(1) resulted in a final refined distance of 0.859(9). The same DFIX command was used in the reanalysis, but this time, the final distance refined to 0.867(10). The experimental section now reads “Hydrogen atom H3 on nitrogen atom N3 was found from a Fourier difference map. Uiso(H) was set to 1.20 of Ueq(N), and the N3-H3 distance was refined using a DFIX command set to 0.87(1) Å [final distance refined to 0.867(10) Å].” to clarify what was done.

Point 4.  On page 5 it is stated that the asymmetric unit contains one cobalt center, but this is really a 1/2 cobalt atom because Z' = 0.5. (It would be helpful if Z' were given along with Z in the experimental details.)

Response 4: The analysis on page 5 has now been edited to say that “The asymmetric unit contains…0.5 cobalt centers”, and Z’ = 0.5 has been listed in the experimental section directly after Z.

Reviewer 3 Report

In this manuscript, Zhang and coworkers report the design and synthesis of a 3-D Co(II) supramolecular framework. The chemical structure was solved by SXRD analysis and was further confirmed by FT-IR, and element analysis. The resultant material was applied for hydroboration and hydrosilylation of ketones and aldehydes. Although the paper is of some interest, it has major flaws. I just listed my concerns below:

  1. As the core idea of this work, the authors designed and synthesized a 3-D Co(II) supramolecular framework for hydrofunctionalization of ketones and aldehydes. However, as indicated by the authors in the manuscript (line 248-252), this catalyst is unstable and becomes homogeneous solution under the catalytic conditions. To be honest, I agree with the authors that this Co-based supramolecular framework would decompose in the presence of the base, KOtBu. And if this is the case, then what is the meaning of synthesizing this material? They could just use homogeneous Co salts-based system to facilitate this transformation. And as control experiments, this also must be done. To make my point clear, I am just questioning the design of this work.
  2. As claimed by the authors, this is a novel 3-D supramolecular structure. However, this work lacks corresponding materials characterizations significantly, like PXRD to prove the crystallinity of the as-prepared catalyst, TEM, SEM to show the morphology of the material, etc.
  3. As a well-studied type of reaction, hydroboration of ketones and aldehydes is proved to be solely catalyzed by the base additive, like potassium carbonate (ACS Omega 2019, 4, 15893−15903). Moreover, boranes and borohydrides have also recently been proven to play a hidden role in hydroboration reactions (ACS Catal. 2020, 10, 13479−13486). I think this should also be discussed in the manuscript.
  4. The amount of self-citation seems exaggerated, especially for an article paper. 18 out of 33 citations are self-citations. I agree that the work and perspectives of the author’s group are undoubtedly important, but I suggest they revise the introduction to be more balanced and include contributions from a diverse set of groups where possible. Please consider both including additional recent examples, but also rephrasing the research description within the introduction to assign credit to a broader set of contributors.

Author Response

Responses to reviewer:

Point 1.  As the core idea of this work, the authors designed and synthesized a 3-D Co(II) supramolecular framework for hydrofunctionalization of ketones and aldehydes. However, as indicated by the authors in the manuscript (line 248-252), this catalyst is unstable and becomes homogeneous solution under the catalytic conditions. To be honest, I agree with the authors that this Co-based supramolecular framework would decompose in the presence of the base, KOtBu. And if this is the case, then what is the meaning of synthesizing this material? They could just use homogeneous Co salts-based system to facilitate this transformation. And as control experiments, this also must be done. To make my point clear, I am just questioning the design of this work.

Response 1: The decomposition of the polymeric materials under the catalytic conditions were unexpected, unlike the reported cobalt-terpyridine coordination polymer (ref. 28). We proposed to design polymeric structures of cobalt with readily accessible ligands containing additional hydrogen-bonding units distinct from terpyridine derivatives and to demonstrate the differences on reactivity and selectivity from previous catalysts, complementing the state-of-the-art of the cobalt-coordination polymer catalysts. According to your suggestion, we performed the reaction using the Co(NCS)2 salt as a precatalyst, and the performance turned out to be much inferior, suggesting the role of the ligand. The results were added in the Table 1 and discussed accordingly. In fact, the homogeneous nature of the catalyst mixture may benefit the isolation and characterization of any reactive intermediate and hence help with the understanding of the catalytic mechanism. We are currently devoted to a detailed understanding of the catalytic process by isolating the soluble species as possible intermediate that promoted the reaction. The results are expected to publish in due course.

Point 2.  As claimed by the authors, this is a novel 3-D supramolecular structure. However, this work lacks corresponding materials characterizations significantly, like PXRD to prove the crystallinity of the as-prepared catalyst, TEM, SEM to show the morphology of the material, etc.

Response 2: As we are in a research group focusing on organic chemistry and catalytic reactions, we have limited capability of conducting more advanced characterization of materials for morphology such as TEM and SEM. However, we were able to collect the PXRD and the results were included and briefly explained.

Point 3.  As a well-studied type of reaction, hydroboration of ketones and aldehydes is proved to be solely catalyzed by the base additive, like potassium carbonate (ACS Omega 2019, 4, 15893−15903). Moreover, boranes and borohydrides have also recently been proven to play a hidden role in hydroboration reactions (ACS Catal. 2020, 10, 13479−13486). I think this should also be discussed in the manuscript.

Response 3: Thanks for the helpful suggestion. We have included a brief discussion on the mentioned work in both the introduction and the catalysis part.

Point 4.  The amount of self-citation seems exaggerated, especially for an article paper. 18 out of 33 citations are self-citations. I agree that the work and perspectives of the author’s group are undoubtedly important, but I suggest they revise the introduction to be more balanced and include contributions from a diverse set of groups where possible. Please consider both including additional recent examples, but also rephrasing the research description within the introduction to assign credit to a broader set of contributors.

Response 4: During the revision, the citations have been revised by citing more work from other groups in the field and removing some less related citations from our group for a more balanced overall citation. Now there are only a few most related citations have remained from this group.

Reviewer 4 Report

The authors report the synthesis and characterization of a supramolecular framework formed by a pyridine-base hydrazide-hydrazone ligand and Co(NCS)2.  The 3D network consists of 2D grid layers connected via hydrogen bonds. The title compound can be employed for hydroboration and also hydrosilylation catalysis for ketones and aldehydes under mild conditions. Chemoselectivity of this catalist system was also observed for aldehyde over ketone as well as ketone over other reducible groups.

The manuscript is well written, however the number of self-citations is too large. I counted 18 times G.Zhang et.al among in sum 33 references. I strongly recommend to modify this reference ratio and give in the introduction also citations of the work of other groups in this area.

On the other hand all references related with the X-ray crystallography section are missing and should be included in the main text body too.

For the chemicals LiNTf2 and HBin should be added also in full name when presented first in text.

Author Response

Responses to reviewer:

Point 1.  The manuscript is well written, however the number of self-citations is too large. I counted 18 times G.Zhang et.al among in sum 33 references. I strongly recommend to modify this reference ratio and give in the introduction also citations of the work of other groups in this area.

Response 1: During the revision, the citations have been revised by citing more work from other groups in the field and removing some less related citations from our group for a more balanced overall citation. Now there are only a few most related citations have remained from this group.

Point 2.  On the other hand all references related with the X-ray crystallography section are missing and should be included in the main text body too.

Response 2: References 36-39 related to the X-ray crystallography are included now in the main text. Thanks for the suggestion.

Point 3.  For the chemicals LiNTf2 and HBin should be added also in full name when presented first in text.

Response 3: This has been fixed now.

Reviewer 5 Report

This manuscript reports the preparation and characterization of a Cobalt supramolecular framework and investigates their catalytic performance hydroboration (or hydrosilylation) reactions of ketones and aldehydes.

The idea of the paper is interesting, and the results are valuable to be published as an original research paper. However, to publish it, some improvements are needed:

 Concerning the characterization of 3D cobalt, please add the FT-IR spectrum of the ligand L and compound 1, and add a sentence commenting on the band shifts upon coordination on the main text. In addition, to assess the thermal stability of 1, TGA should be included.

 Please verify if the XRPD pattern of the as-synthesized product closely matches the simulated one from the single-crystal data. If so, please include this comparison in the supplementary information and add a sentence in the main text.

 Please check the document, and confirm that when you refer to 1, 1 is in bold.

 Please add a column with the data for 1D Cutpy data for comparison in Table 1 and Table 2.

Author Response

Responses to reviewer:

Point 1.   Concerning the characterization of 3D cobalt, please add the FT-IR spectrum of the ligand L and compound 1, and add a sentence commenting on the band shifts upon coordination on the main text. In addition, to assess the thermal stability of 1, TGA should be included.

Response 1: Thanks for the helpful comments. We have added the FT-IR spectrum of the ligand and included a comparison figure for both the ligand and compound 1 in the SI. A brief discussion on the IR is now added in the main text. Unfortunately, we are not capable of conducting the thermal analysis (TGA) of the compound for the moment.

Point 2.   Please verify if the XRPD pattern of the as-synthesized product closely matches the simulated one from the single-crystal data. If so, please include this comparison in the supplementary information and add a sentence in the main text.

Response 2: According to the suggestion, we have collected the PXRD and the results were included and briefly explained. The mismatch of the PXRD with the simulated XRD pattern from single crystal diffraction may suggest a phase transition due to the rapid loss of volatile solvents from the large voids in the network. However, the poor crystallinity of the dried sample prevented us from determining the actual structural change.

Point 3.   check the document, and confirm that when you refer to 1, 1 is in bold.

Response 3: This has been fixed now.

Point 4.    Please add a column with the data for 1D Cutpy data for comparison in Table 1 and Table 2.

Response 4: The data for 1D Co-tpy has been added into Table 1. For Table 2, since the catalytic reactions presented here are not exactly the same as reported with 1D Co-tpy, we were unable to add a new column to make a complete comparison. Nevertheless, with appropriate citations, the readers could easily track the previous work for the purpose of comparison.

Round 2

Reviewer 3 Report

It is pleasing to see that the authors have addressed the questions raised in the first reviewing process. The paper is now ready for publication on Chemistry.